# Financial Performance Gaps and Corporate Social Responsibility

**Xinming Deng and Xianyi Long ***

School of Economics and Management, Wuhan University, Wuhan 430072, China; xmdeng@whu.edu.cn
* Correspondence: longxianyi@whu.edu.cn; Tel.: +0086-27-68753213

**Abstract:** Based on the behavioral theory of firm and prospect theory, we investigate how corporate social responsibility (CSR) activities will respond to underperformance in past and in future. Using samples of Chinese listed firms from 2011 to 2016, this paper found that CSR increases with the distance by which financial performance in the last year falls below goals and decreases with the distance by which expected financial performance will fall below targets. In addition, the future underperformance will weaken the effect of the past underperformance on CSR. Besides, the value of financial performance in the last year will weaken the impact of underperformance in the last year on CSR and strengthen the impact of underperformance in the next year on CSR. The findings suggest that future studies should take both value of financial performance and performance gaps into consideration to have a better understanding of organizational decisions and behaviors.

**Keywords:** backward-looking performance gap; forward-looking performance gap; the value of financial performance; CSR

## 1. Introduction

Studies on how corporate social responsibility (CSR) is shaped by financial performance have failed to gain consensus. Some studies have suggested that they could be positively related since financial performance is a driver of CSR activities [1]. On the contrary, other studies have argued that firms could gain legitimacy and resources through CSR activities [2,3], thus firms experiencing poor financial performance are likely to engage in CSR activities. Since there are competing mechanisms, some studies have found that financial performance is not related with CSR activities [4]. We think the way firms interpret their financial performance can attribute to such mixed findings: it is the gap between financial performance and some goals or targets, rather than the value of financial performance itself, that determines CSR activities.

The behavioral theory of firm may provide a promising lens to address such inconsistency. Based on bounded rationality assumptions, organizations will set goals or targets to simplify their evaluation of financial performance [5]. Prior studies have investigated the impacts of the performance gap between financial performance and such goals or targets on research and development (R and D) search intensity [6], strategy change [7], mergers and acquisitions [8], diversification [9], and internalization [10]. It is suggested by the behavioral theory of firm that there are two search models in the decision-making process: backward-looking and forward-looking [6]. The backward-looking model suggests that firms could accumulate wisdom and experience from prior choices and make adjustments to their later decisions based on performance feedback [11]. Firms will have the motivation to search for alternative plans if they have failed to reach corporate goals. In contrast, the forward-looking model suggest that firms will evaluate the possible outcomes of engaging specific planned behaviors, and likely losses in future will encourage firms to search for alternative plans [12] to attain corporate targets. It is suggested that the right extent of CSR could benefit firms by the way of corporate reputation [13],

consumers' evaluation and loyalty [14–16], stakeholder relations [17], financial performance [18], attractiveness to institutional investors [19], firm capabilities, such as operational efficiencies [20], and other positive employee responses, such as organizational citizenship behavior [21]. In this way, CSR could be used as an effective alternative choice if firms have failed to reach their financial performance goals or will fail to attain their targets in the future.

However, past and future underperformance may have different meanings to firms. Prospect theory suggests that loss in the past will make people more risk-averse and likely loss in the future will make people more risk-seeking [22]. Since CSR activities usually involve lower risk-taking than R and D or internationalization behaviors, we assume that underperformance in the past will increase CSR activities and underperformance in the future will decrease such practices. Besides, we are interested in whether the value of financial performance will influence the impacts of different performance gaps on CSR activities. Chinese listed firms trading on Shenzhen or Shanghai stock exchanges during the period from 2011 to 2016 will be chosen as samples to test the aforementioned arguments. It is complementary to choose China as a research setting since most prior studies about performance gaps or CSR have chosen developed countries as contexts, and the impacts of CSR activities are indeed different in developing and developed countries [23].

This paper may shed new light on the literature about CSR, the behavioral theory of firm, as well as organizational risk taking. First, this paper contributes to the literature on CSR by providing the behavioral theory of firm as a new theory explanation. Prior studies have found many important determinants of CSR activities such as the firm's size, female directors, institutions, and competition intensity [2,4,24–27], but have failed to reach consensus on how CSR activities are influenced by financial performance. Thus, we can have a better understanding about CSR activities from the performance gap perspective. Second, we contribute to the behavioral theory of firm and organizational risk-taking by providing CSR as an alternative plan in the past underperformance situation. Prior studies have explored the influence of performance gaps on organizational behaviors, mostly in the form of R and D activities which are high risk-seeking [6]. CSR activities are usually a lower risk burden on organizations. Thus, this paper aims to understand the impacts of performance gaps on organization behaviors from a different perspective. Finally, we contribute to the behavioral theory of firm by incorporating the value of financial performance and performance gaps into a more comprehensive framework. Based on bounded rationality, organizations would set some goals and targets to evaluate their financial performance [5], but it does not guarantee that the value of financial performance itself doesn't matter. By investigating the moderating role of the value of financial performance, we can better interpret the impacts of financial performance on organizational behaviors.

## 2. Theoretical Background and Hypotheses

### 2.1. Backward-looking Search and Corporate Social Responsibility (CSR)

The backward-looking search model was first described by the behavioral theory of firm by Cyert and March (1963). They assumed that organizational behaviors are history-dependent, goal-directed, and conducted by simple rules [5]. Firms are goal-directed and are prone to set some goals to conduct their decisions and behaviors. Firms are history-dependent and they will adjust their goals based on their evaluation of performance history. That is to say, firms will stick to their routines if past performance has reached their goals and switch to search for other alternative plans if past performance has failed to reach their goals [28]. This backward-looking logic has been widely reported in the literature on organizational risk taking [29], strategic change [7], and R and D actions [30].

Even though there are many studies about the impacts of CSR practices on financial performance [31], studies about the impacts of financial performance on CSR practices are inconclusive. The "available resource hypothesis" proposes that they are positively related since CSR practices could burden costs to firms [32] and good financial performance guarantees enough resources that could be allocated to CSR practices [1]. However, the "social exchange hypothesis" assumes that

firms will increase CSR practices under poor financial performance conditions since CSR practices could play an important role to enhance organization legitimacy and exchange for key resources such as confidence of investors [33], satisfaction of employees [34], loyalty of customers [35], and good relationship with governments [36]. These two contradictory explanations could be attributed to the failure to distinguish different motivations behind CSR practices. The good-performance-based CSR practices may be motivated by corporate resources and generosity, and poor-performance-based CSR practices may be conducted to solve problems. To distinguish these two different motivations, we could use the backward-looking model from the behavioral theory of firm.

It is suggested that firms will evaluate and interpret their past performance against some goals to make decisions [37]. If past financial performance has reached their goals, corporate social responsibility is more likely to be motivated by firms' generosity, and if past financial performance has failed to reach their goals, corporate social responsibility is more likely to be conducted to solve problems and due to a wish to return to goals in the next evaluation round (generally next year). These two different patterns have been categorized as problem-driven and slack-motivated search when investigating organizational R and D search [6].

Besides, it is suggested by the behavioral theory of firm that firms are goal-directed, history-dependent and conducted by simple rules [37]. Firms will set simple, objective, and measurable goals to provide a reference point. These goals are usually connected to financial performance [38]. It is the gap between past performance and these goals that help firms to make judgements about how well they have done in the past [39]. Firms usually take financial performance from the past few years to set goals since environments in the long-term period are likely to be non-stationary and predictions are likely to be biased. Financial performance closer to the last year will be given higher weights.

It has been confirmed that the framing of an outcome would change subsequent levels of risk acceptance, and losses in the past will make people and organizations more risk-averse [22]. Prior findings that underperformance in the past will make firms invest higher in R and D activities [6] have failed to take organizational risk preference into consideration. R and D activities are more risk-laden than CSR practices. When facing underperformance in the last year and the choice between R and D activities and CSR practices, if possible, firms are more likely to choose CSR activities.

To summarize, firms will set simple goals dependent on a weighed combination of financial performance in past few years. Then, the gap between financial performance and these goals determines firms' decisions about CSR practices: the higher the extent to which financial performance falls below goals leads to higher CSR. We will not make a proposition about CSR motivated by firms' generosity since it is hard to determine how generous a firm is.

**Hypothesis 1:** *CSR increases with the distance of past performance below goals*.

*2.2. Forward-looking Search and CSR*

Based on prospect theory, Chen (2008) has developed the forward-looking search model of R and D investments. This study argued that technology development has armed firms with higher information processing and computation capability, which then enables firms to have a better understanding of future. Thus, many studies have attempted to investigate the effect of firms' understanding of potential future [40]. This forward-looking perspective assumes that firms will have a cognition of environments and consider all possible outcomes of different organizational behaviors before they decide which plan to engage [12], thus, they have expectations of corporate performance in the future. In this way, firms are more likely to choose those plans which could reach their performance targets.

However, there may be two alternative choices that could help firms to reach future targets: R and D activities and CSR practices. How will firms choose? We think prospect theory could provide us with a useful framework. It is suggested by prospect theory that people are risk-averse if they are likely to gain in the future and risk-seeking if they are likely to lose in the future [22]. Management literature

has extended this logic from individuals to organizations and found that organizations are risk-averse when they are likely to reach or exceed their targets and risk-seeking when they are unlikely to reach organizational targets [41]. From the literature on R and D activities, we see that R and D activities are high in risk-taking behaviors and that their failure rate could be up to 80%, and successful R and D programs still need time to be translated into profits to firms. In contrast, CSR practices would burden little risk to firms and can be translated into competition advantage both in the short-term by means of resources exchange [33,35], and in the long-term by means of reputation [42]. Therefore, firms are less likely to choose CSR practices if they are likely to perform below targets. The results of Chen (2008) show that firms' R and D activities will increase if firms are less likely to reach performance targets, and this could lend support to our arguments.

**Hypothesis 2:** *CSR will decrease with the distance by which expected performance falls below targets.*

*2.3. Interactions of Backward- and Forward-looking Search*

In the backward-looking model, underperformance in the last year will make firms more likely to be risk-averse and thus choose to have more CSR practices. In the forward-looking model, likely underperformance will make firms more likely to be risk-seeking and thus have less CSR practices. It is reasonable to argue that both the backward- and forward-looking perspectives will be taken into consideration in the decision-making process. Therefore, we are going to investigate whether the interaction of these two different performance gaps will influence firms' CSR.

According to hypothesis 1, firms will choose to increase CSR practices if they have experienced underperformance in the last year. On the contrary, hypothesis 2 posits that firms will choose to decrease CSR practices if they will experience underperformance in the next year. What will firms do if they have experienced underperformance in the last year and will experience underperformance again in the next year? We argue that such a desperate situation will make firms more risk-seeking [6] since it could not be worse off, and thus, firms will invest more in risk-seeking behaviors such as R and D activities.

**Hypothesis 3:** *The performance gap against targets in the next year will weaken the effect of the performance gap in the last year on CSR.*

*2.4. Moderating Effect of the Value of Financial Performance*

Based on behavioral theory of firm and prospect theory, above we have made propositions on the relationship between underperformance and CSR. However, this does not mean that the value of financial performance alone does not make sense to firms. A financial performance may be satisfactory even though it failed to reach the goals, or it will fall below targets, especially when the performance gap is small and acceptable. Attribution theory may be helpful to explain how the value of financial performance will coexist with performance gaps.

Attribution theory suggests that there is an "actor–observer effect" that we tend to attribute others' failure to internal reasons and our failure to external reasons [43]. Following this logic, firms that have experienced underperformance feedback or will fail to reach targets are likely to attribute underperformance to external reasons. One potential external reason may be that the goals or targets are inappropriately set and thus difficult to reach. This kind of attribution is likely to make sense when financial performance alone is good enough. In this way, a higher value of financial performance will weaken the effect of underperformance gaps.

Besides, a higher value of financial performance could also increase firms' tolerance to risk. "Self-serving attribution bias" indicates that we tend to take responsibility for good outcomes and deny responsibility to poor outcomes [44]. A higher value of financial performance is usually positively

connected to managers' confidence under the existence of the "self-serving attribution bias," and managers will take responsibility of firms' success [45]. It is suggested that managers' confidence or overconfidence could result in high risk-taking decisions such as R and D actions [46]. In summary, a higher value of financial performance means more available resources and firms are prone to take more risks.

Hypothesis 1 suggests that CSR will increase with the distance by which past performance falls below goals since firms have motivations to search for alternative plans and are highly risk-averse at the same time. But the higher value of financial performance will decrease firms' motivation to search for alternative plans and increase firms' tolerance to risk. Therefore, the relationship between the distance by which past performance falls below goals and CSR will be weakened. Hypothesis 2 suggests that CSR will decrease with the distance by which expected performance will fall below targets since firms are risk-seeking and will search for other risk-taking investments such as R and D actions. Then, a higher value of financial performance will lead firms to be less likely to invest in low-risk practices such as CSR practices. Thus, we have made the following propositions:

**Hypothesis 4:** *The value of financial performance will weaken the effect of the distance by which past performance falls below goals on CSR.*

**Hypothesis 5:** *The value of financial performance will strengthen the effect of the distance by which expected performance will fall below targets on CSR.*

## 3. Method

### 3.1. Samples

We chose all traded firms listed on either Shanghai Stock Exchange or Shenzhen Stock Exchange during the period from 2011 to 2016 as samples. In order to reduce the possibility of reverse causality, all control and explanatory variables are lagged by one additional year [3]. Since some firms conducted initial public offering (IPO) during this period, our sample constitutes an imbalanced panel dataset.

We compiled a comprehensive dataset from the China Stock Market and Accounting Research (CSMAR) and HEXUN website. The CSMAR was developed by a leading global provider of Chinese financial market data. Most data used in this study either come directly from, or are computed based on the original data from, the CSMAR database. This database provides information about firms' accounting and governance, as well as analysts' estimates for all Chinese firms that are listed on either the Shanghai Stock Exchange or the Shenzhen Stock Exchange. This database enabled us to construct corporate financial performance goals and targets which are core variables in this study. We used the HEXUN website as a source to measure corporate social responsibility of Chinese listed firms. HEXUN is the leading website providing financial and securities information service in China, and it is the only database in China that provides corporate social responsibility information of all listed firms. Ranking CSR Ratings (RKS) is widely used as database for CSR researches, but it only concludes listed firms who have disclosed CSR reports and thus cover only small part of Chinese listed firms. For instance, only 747 Chinese listed firms disclosed CSR report but there are more than 3590 listed firms trading on Shenzhen or Shanghai Stock Exchange in 2016. CSR activities on HEXUN are described by 37 indices and categorized into five types: shareholder responsibility, employee responsibility, suppliers and customers responsibility, environment responsibility, and society responsibility. Information of these activities was collected from the CSR report and the annual report. This database has been increasingly used as a main data source by Chinese studies or studies about CSR of Chinese listed firms [47,48].

In order to make the results reliable, we processed the data as follows. First, we excluded observations labeled as special treatment (ST) since these firms are in abnormal conditions. Second, we excluded samples in restricted industries such as the banking industry, the security industry, and the insurance industry. Third, samples that were listed on the stock exchange for less than four years

were excluded. Fourth, we excluded samples with missing key information. As a result, we got a final sample of 10,280 firm-year observations. To account for any selection bias, we used a two-stage Heckman model by running a panel logit model on the entire raw sample and then incorporated the inverse Mills ratio calculated from the first-stage model into the second-stage model, which only include our final dataset.

*3.2. Models*

We estimated a linear relationship between financial performance gaps and CSR activities. This model will contain a dependent variable, two explanatory variables, three interactions of explanatory variables and moderators, as well as all other control variables (see Equation (1)). Following Greve (2003) and Chen (2008), we incorporated an indicator variable to distinguish outperformance and underperformance samples in the last year. $I_1$ is the indicator variable for underperformance samples that is equal to 1 if past performance falls below the goals (firm-specific or industry-specific). And $1-I_1$ is an indicator for outperformance firms that is equal to 1 if past performance exceeds goals (firm-specific or industry-specific). Similarly, $I_2$ is the indicator variable for likely underperformance samples in the next year that is equal to 1 if the firms' expected performance is likely to fall below its performance targets (firm-specific or industry-specific). And $1-I_1$ is an indicator for likely outperformance samples in the next year that is equal to 1 if the firms' expected performance is likely to exceed its performance targets (firm-specific or industry-specific).

$$
\begin{aligned}
CSR_{i,t} = \ & \beta_0 CSR_{i,t-1} + \beta_1 I_1 (P_{i,t-1} - A_{i,t-1}) + \beta_2 I_2 (EP_{i,t+1} - T_{i,t}) \\
& + \beta_3 I_1 (P_{i,t-1} - A_{i,t-1}) I_2 (EP_{i,t+1} - T_{i,t}) + \beta_4 ROA_{i,t-1} I_1 (P_{i,t-1} - A_{i,t-1}) \\
& + \beta_5 ROA_{i,t-1} I_2 (EP_{i,t+1} - T_{i,t}) + \beta_6 C_{i,t-1} + \beta_6 CSR_{ind,t} \\
& + \beta_7 M_{i,t} + \beta_8 YEAR + \varepsilon_{i,t-1}
\end{aligned}
\tag{1}
$$

In this model, $CSR_{i,t}$ is designated as firm i's corporate social responsibility in period t. We include $CSR_{i,t-1}$ as a control to reduce the influence of autocorrelation [49,50]. $P_{i,t-1}$ is a measure of a firms' financial performance at period $t-1$ and $A_{i,t-1}$ is a measure of firm i's performance goals for period $t-1$. $EP_{i,t+1}$ is a measure of a firm's expected financial performance at period $t+1$ and $T_{i,t}$ is a measure of firm i's performance targets for period $t$. $C_{i,t-1}$ is a group of one-year lagged control variables that would influence corporate social responsibility. $CSR_{ind,t}$ is used to control the industry average level of CSR at the same period since we are going to conduct cross-industry research and CSR practices may perform in different domains and at different levels across industries [51,52]. $M_{i,t}$ is the inverse Mills ratio calculated from the first stage panel logit model using all raw samples. We also include five dummies to control for time effect in our model. Ordinary Least Square (OLS) regression was used to test our model with penal data. Specifically, we used the fixed procedure in STATA program which was advised by the Hausman test and reported the robust standard errors.

*3.3. Variables*

3.3.1. Dependent Variables

*Corporate social responsibility (CSR).* In line with arguments of prior studies [31,53], we think corporate responsible behaviors to all stakeholders should be included to measure CSR since they could benefit firms in different ways. HEXUN website provides total score of corporate social responsibilities to these stakeholders: shareholders, employees, suppliers and customers, environment, and society. So we use this total score to measure CSR. This variable has 100 as the highest value and the lowest value could be negative since some firms have been punished for environmental pollution or unsafe working environment.

### 3.3.2. Independent Variables

*Past performance.* There are many ways to evaluate corporate financial performance [54] and we choose ROA in this paper for its ability to avoid distortions by differences in leverages across firms and industries [55]. This variable is measured as ROA in the last year.

*Performance goals.* In line with prior studies [56], we computed firms' performance goals using the exponentially weighed moving average of past performance (see Equation (2)). In such a situation, performance goals are adapting more to the closer past performance to avoid bias. In this equation, we tried different weights for ($a_1$) from 0 to 1 by increments of 0.1 and found consistent results. So, we report the results based on $a_1 = 0.4$ which has the highest value of $R^2$. In that way, firms' performance goals for $t−1$ is the weighted combination of focal firms' performance at $t−2$ (with a weight of 0.6) and performance at $t−3$ (with a weight of 0.4). The industry-specific performance goals in $t−1$ are a combination of the median performance of firms in the same industry at $t−2$ (with a weight of 0.6) and at $t−3$ (with a weight of 0.4).

$$A_{i,t−1} = (1 − a_1)P_{i,t−2} + a_1 A_{i,t−2} \tag{2}$$

*Expected performance.* Following the measurement of prior studies [6], we multiplied the median of all analysts' one-year prediction for firms' earning per share by the total number of shares to capture the forecasted earnings and then divided by total assets to capture expected performance. It has been confirmed that there are continuous communications between CEOs and analysts and there is high correlation between executives' forecasts and analysts' forecasts [55].

*Targets.* Based on the logic of the behavioral theory of firm, performance targets are set on the understanding of historical performance [37]. In that way, we measured corporate financial performance targets by regressing on firms' ROA from time $t−1$ to $t−3$ and then predicted the value of ROA at time t.

*Moderators.* We had two moderators in this paper: the performance gap between expected performance and targets and the value of financial performance. The former could be computed by using expected performance at period $t+1$ minus performance targets at period t, both of which have been discussed above. The latter could be measured by past performance which also has been discussed above as an explanatory variable.

*Control variables.* We controlled for many factors that may influence CSR. At the firm level, we control for firm age, firm size, boardroom gender diversity, subsidy, ownership type, visibility, political ties, slack resources, and R and D investment. Firm age is the number of years since a firm's foundation. We use its natural logarithm to correct for its skewness. Firm size was measured by the natural logarithm of firms' total assets. Boardroom gender diversity is the proportion of female directors. Studies have shown that firms with high boardroom gender diversity are likely to invest more in CSR activities [57]. Subsidy is reported in firms' annual reports and we use its natural logarithm to correct for its skewedness. Ownership type was a dummy variable without state-owned enterprises (non-SOEs) coded as 0 and state-owned enterprises (SOEs) coded as 1. Visibility is the percentage of sales, general and administrative expenses to total sales. Higher visibility makes the information of firms more transparent to outsiders [58] and provides higher motivation for firms to solve problems timely. Political ties were measured as Board Chairman or CEO's political membership. Firms with their Board Chairman or CEO acting as a deputy of the People's Congress or a member of the Chinese People's Political Consultative Conference (CPPCC) are coded as 1, and otherwise 0 [59]. We measured slack resources by taking the total net cash-flow from a firm's operations, financing, and investing activities, scaled by its total sales. We also controlled for R and D investment to account for firms' trade-off between R and D activities and CSR activities, which is measured by proportion of R and D expenditure on sales. At the industry level, we controlled for the median of CSR of all firms in the same industry since CSR is likely to be influenced by mimetic isomorphism pressures [60]. At the time level, we included five dummies to control for the microenvironment effect. We also included a one-year lagged CSR to control for the effect of organizational routines.

## 4. Results

### 4.1. Descriptive Statistics

Table 1 provides the descriptive statistics for our final dataset, which consists of 10,280 firm-year observations. The mean of the CSR is 28.77, and the median is 22.31, which indicates that Chinese listed firms' social performance is still at low levels. Table 2 presents the correlation matrix of all variables. The correlations between firm-specific and industry-specific past underperformance gaps (0.754) as well as firm-specific and industry-specific future underperformance gaps (0.632) are not high, so that we have to report their effects on CSR separately. The negative correlation between R and D investment and CSR activities ($\beta = -0.064$, $p < 0.01$) indicates that firms may make trade-off between these two alternative plans in an underperformance situation.

### 4.2. Main Effect

Table 3 reports the results of the main effect as well as the joint effect and interactive effect between two independent variables. And Table 4 shows the moderating effect of the value of financial performance. To avoid multicollinearity, we centered all of the main explanatory variables and moderators that were used to create interaction terms [61]. From Tables 3 and 4, it is surprising to find that the coefficients of CSR at time $t-1$ are low and significant, which may be contradictory to prior studies that suggest CSR is highly institutionalized and becomes part of organizational routines [62]. On the contrary, it provides support to our argument that CSR is an alternative choice to solving problems which implicitly suggests that CSR practices are not organizational routines. Besides, the coefficient of median of industry CSR is moderate and significant which supports the mimetic isomorphism tendency of CSR [60].

**Table 1.** Descriptive Statistics for Panel Data.

| Variable | Mean | Median | Std. Dev. | Min | Max |
|---|---|---|---|---|---|
| $CSR_t$ | 28.77178 | 22.31 | 19.59721 | −3.23 | 76.79 |
| $I_1(P_{t-1,firm} - A_{t-1,firm}) < 0$ | −0.003930 | −0.00403 | 0.0081349 | −0.0333403 | 0 |
| $(1 - I_1)(P_{t-1,firm} - A_{t-1,firm}) \geq 0$ | 0.046041 | 0 | 0.0617546 | 0 | 0.3408224 |
| $I_1(P_{t-1,ind} - A_{t-1,ind}) < 0$ | −0.003838 | −0.00364 | 0.0087479 | −0.0436898 | 0 |
| $(1 - I_1)(P_{t-1,ind} - A_{t-1,ind}) \geq 0$ | 0.045483 | 0 | 0.060839 | 0 | 0.3379822 |
| $I_2(EP_{t+1,firm} - T_{t,firm}) < 0$ | −0.01797 | 0 | 0.0330775 | −0.2546664 | 0 |
| $(1 - I_2)(EP_{t+1,firm} - T_{t,firm}) \geq 0$ | 0.011615 | 0.025082 | 0.0292051 | 0 | 0.236933 |
| $I_2(EP_{t+1,ind} - T_{t,ind}) < 0$ | −0.017181 | 0 | 0.0316404 | −0.2341704 | 0 |
| $(1 - I_2)(EP_{t+1,ind} - T_{t,ind}) \geq 0$ | 0.019601 | 0.025264 | 0.0339168 | 0 | 0.1887244 |
| $CSR_{t-1}$ | 28.79537 | 22.33 | 19.33354 | −3.16 | 77.17 |
| Median of Industry $CSR_t$ | 22.82581 | 21.45 | 4.322903 | 16.68 | 45.6 |
| Boardroom gender diversity$_{t-1}$ | 0.125327 | 0.111111 | 0.1125708 | 0 | 0.4545455 |
| Subsidy$_{t-1}$ | 6.733886 | 6.957299 | 1.603568 | 0 | 8.828683 |
| Ownership type$_{t-1}$ | 0.496352 | 1 | 0.500011 | 0 | 1 |
| Visibility$_{t-1}$ | 0.097298 | 0.078904 | 0.0702845 | 0.0103887 | 0.3871848 |
| ROA$_{t-1}$ | 0.045202 | 0.041784 | 0.0544508 | −0.174401 | 0.240641 |
| Firm age$_{t-1}$ | 1.216337 | 1.230449 | 0.124859 | 0.845098 | 1.518514 |
| Firm size$_{t-1}$ | 1.025287 | 1.019734 | 0.0223528 | 0.9710937 | 1.088251 |
| Political ties$_{t-1}$ | 0.256445 | 0 | 0.4366917 | 0 | 1 |
| Slack resources$_{t-1}$ | 0.009901 | 0.003054 | 0.2671814 | −1.062666 | 1.552143 |
| R and D investment$_{t-1}$ | 0.026242 | 0.0113 | 0.0383173 | 0 | 0.2236707 |

Note: ROA means return on assets.

**Table 2.** Correlation Matrix ($N$ = 10,280).

| Variable | 1 | 2 | 3 | 4 | 5 | 6 | 7 | 8 | 9 | 10 |
|---|---|---|---|---|---|---|---|---|---|---|
| $CSR_t$ | | | | | | | | | | |
| $I_1(P_{t-1,firm} - A_{t-1,firm}) < 0$ | 0.133 *** | | | | | | | | | |
| $(1-I_1)(P_{t-1,firm} - A_{t-1,firm}) \geq 0$ | 0.033 * | 0.356 *** | | | | | | | | |
| $I_1(P_{t-1,ind} - A_{t-1,ind}) < 0$ | 0.123 *** | 0.754 *** | 0.317 *** | | | | | | | |
| $(1-I_1)(P_{t-1,ind} - A_{t-1,ind}) \geq 0$ | 0.025 | 0.351 *** | 0.983 *** | 0.324 *** | | | | | | |
| $I_2(EP_{t+1,firm} - T_{t,firm}) < 0$ | 0.098 *** | −0.006 | −0.056 *** | 0.03 | −0.037 ** | | | | | |
| $(1-I_2)(EP_{t+1,firm} - T_{t,firm}) \geq 0$ | −0.076 *** | 0.049 *** | 0.187 *** | 0.047 *** | 0.197 *** | 0.206 *** | | | | |
| $I_2(EP_{t+1,ind} - T_{t,ind}) < 0$ | 0.237 *** | 0.187 *** | 0.086 *** | 0.318 *** | 0.113 *** | 0.632 *** | 0.114 *** | | | |
| $(1-I_2)(EP_{t+1,ind} - T_{t,ind}) \geq 0$ | 0.192 *** | 0.227 *** | 0.503 *** | 0.227 *** | 0.536 *** | 0.131 *** | 0.402 *** | 0.298 *** | | |
| $CSR_{t-1}$ | 0.668 *** | 0.094 *** | 0.029 | 0.102 *** | 0.023 | 0.178 *** | −0.056 *** | 0.348 *** | 0.208 *** | |
| Median of Industry $CSR_t$ | 0.218 *** | 0.01 | −0.075 *** | −0.011 | −0.095 *** | 0.033 ** | −0.017 | 0.066 *** | −0.019 | 0.179 *** |
| Boardroom gender diversity$_{t-1}$ | −0.073 *** | 0.064 *** | 0.066 *** | 0.039 ** | 0.065 *** | −0.011 | −0.001 | 0.006 | 0.017 | −0.080 *** |
| Subsidy$_{t-1}$ | 0.113 *** | −0.029 | −0.064 *** | −0.014 | −0.069 *** | 0.073 *** | −0.110 *** | 0.102 *** | 0.002 | 0.129 *** |
| Ownership type$_{t-1}$ | −0.142 *** | 0.199 *** | 0.239 *** | 0.145 *** | 0.234 *** | −0.066 *** | 0.009 | 0.034 ** | 0.077 *** | −0.150 *** |
| Visibility$_{t-1}$ | −0.019 | 0.126 *** | 0.218 *** | 0.027 | 0.176 *** | −0.070 *** | 0.100 *** | −0.093 *** | 0.163 *** | −0.041 *** |
| ROA$_{t-1}$ | 0.269 *** | 0.321 *** | 0.436 *** | 0.287 *** | 0.427 *** | 0.450 *** | 0.298 *** | 0.757 *** | 0.764 *** | 0.368 *** |
| Firm age$_{t-1}$ | −0.008 | −0.056 *** | −0.068 *** | −0.035 * | −0.053 *** | −0.006 | 0.039 *** | −0.042 *** | −0.031 | −0.031 * |
| Firm size$_{t-1}$ | 0.361 *** | −0.216 *** | −0.293 *** | −0.065 *** | −0.267 *** | 0.141 *** | −0.161 *** | 0.171 *** | −0.021 | 0.391 *** |
| Political ties$_{t-1}$ | 0.044 *** | 0.079 *** | 0.049 *** | 0.071 *** | 0.046 *** | 0.02 | −0.054 *** | 0.072 *** | 0.029 | 0.062 *** |
| Slack resources$_{t-1}$ | 0.044 *** | −0.007 | 0.011 | 0.004 | 0.014 | 0.100 *** | 0.119 *** | 0.069 *** | 0.085 *** | 0.032 * |
| R and D investment$_{t-1}$ | −0.071 *** | 0.134 *** | 0.176 *** | 0.098 *** | 0.160 *** | −0.064 *** | −0.081 *** | 0.007 | 0.033 * | −0.064 *** |

| | 11 | 12 | 13 | 14 | 15 | 16 | 17 | 18 | 19 | 20 |
|---|---|---|---|---|---|---|---|---|---|---|
| Median of Industry $CSR_t$ | | | | | | | | | | |
| Boardroom gender diversity$_{t-1}$ | −0.050 *** | | | | | | | | | |
| Subsidy$_{t-1}$ | −0.245 *** | −0.046 *** | | | | | | | | |
| Ownership type$_{t-1}$ | −0.163 *** | 0.158 *** | −0.052 *** | | | | | | | |
| Visibility$_{t-1}$ | −0.199 *** | 0.036 *** | 0.087 *** | 0.058 *** | | | | | | |
| ROA$_{t-1}$ | 0.039 *** | 0.028 | 0.082 *** | 0.124 *** | 0.126 *** | | | | | |
| Firm age$_{t-1}$ | 0.049 *** | 0.024 | −0.109 *** | −0.139 *** | 0.043 *** | −0.109 *** | | | | |
| Firm size$_{t-1}$ | 0.183 *** | −0.146 *** | 0.328 *** | −0.351 *** | −0.218 *** | 0.005 | 0.085 *** | | | |
| Political ties$_{t-1}$ | −0.079 *** | 0.052 *** | 0.062 *** | 0.238 *** | 0.018 | 0.078 *** | −0.066 *** | 0.002 | | |
| Slack resources$_{t-1}$ | 0.074 *** | −0.000 | −0.032* | −0.029 | −0.025 | 0.091 *** | 0.070 *** | 0.065 *** | −0.026 | |
| R and D investment$_{t-1}$ | −0.179 *** | 0.02 | 0.146 *** | 0.273 *** | 0.132 *** | 0.101 *** | −0.163 *** | −0.237 *** | 0.021 | −0.078 *** |

Notes: * $p < 0.1$, ** $p < 0.05$, *** $p < 0.01$.

**Table 3.** Results of Main Effects.

| | Baseline Model | Model 1 (Backward-Looking) | | Model 2 (Forward-Looking) | | Model 3 (Joint Model) | | Model 4 (Interaction Model) | |
|---|---|---|---|---|---|---|---|---|---|
| | Firm | Firm | Industry | Firm | Industry | Firm | Industry | Firm | Industry |
| $CSR_{t-1}$ | 0.0756 *** | 0.0739 *** | 0.0800 *** | 0.0691 *** | 0.0731 *** | 0.0679 *** | 0.0774 *** | 0.0699 *** | 0.0777 *** |
| | (0.0160) | (0.0162) | (0.0161) | (0.0159) | (0.0160) | (0.0161) | (0.0161) | (0.0161) | (0.0161) |
| Median of Industry $CSR_t$ | 0.485 *** | 0.480 *** | 0.466 *** | 0.450 *** | 0.475 *** | 0.447 *** | 0.452 *** | 0.449 *** | 0.453 *** |
| | (0.0711) | (0.0708) | (0.0703) | (0.0702) | (0.0709) | (0.0701) | (0.0699) | (0.0701) | (0.0698) |
| Boardroom gender diversity$_{t-1}$ | −4.542 | −4.826 † | −4.526 | −4.376 | −4.719 † | −4.613 | −4.667 | −4.491 | −4.746† |
| | (2.845) | (2.842) | (2.848) | (2.842) | (2.844) | (2.841) | (2.845) | (2.832) | (2.840) |
| Subsidy$_{t-1}$ | 0.0523 | 0.0419 | 0.0622 | 0.0548 | 0.0586 | 0.0459 | 0.0663 | 0.0497 | 0.0733 |
| | (0.134) | (0.133) | (0.134) | (0.133) | (0.134) | (0.133) | (0.134) | (0.131) | (0.134) |
| Ownership type$_{t-1}$ | −1.818 | −2.045 | −1.870 | −2.056 † | −1.928 | −2.237 † | −1.978 | −2.410 † | −2.128† |
| | (1.235) | (1.229) | (1.240) | (1.232) | (1.232) | (1.229) | (1.236) | (1.230) | (1.233) |
| Visibility$_{t-1}$ | 18.30 ** | 17.76 ** | 14.01 * | 17.87 ** | 18.00 ** | 17.43 ** | 13.47 * | 17.73 ** | 13.95 * |
| | (6.846) | (6.802) | (6.776) | (6.763) | (6.804) | (6.739) | (6.753) | (6.702) | (6.740) |
| $ROA_{t-1}$ | 21.59 *** | 35.62 *** | 55.29 ** | 20.18 *** | 19.08 *** | 31.93 *** | 60.68 ** | 32.78 *** | 60.44 ** |
| | (4.853) | (7.166) | (20.37) | (4.813) | (4.835) | (7.132) | (20.34) | (7.081) | (20.33) |
| Firm age$_{t-1}$ | −16.64 | −13.44 | −15.10 | −14.25 | −15.35 | −11.62 | −13.11 | −11.45 | −13.08 |
| | (13.20) | (13.29) | (13.24) | (13.05) | (13.12) | (13.14) | (13.14) | (13.12) | (13.15) |
| Firm size$_{t-1}$ | 180.7 *** | 174.8 *** | 185.1 *** | 184.8 *** | 185.4 *** | 179.7 *** | 189.3 *** | 177.4 *** | 187.9 *** |
| | (29.36) | (29.58) | (29.60) | (29.06) | (29.34) | (29.30) | (29.55) | (29.16) | (29.45) |
| Political ties$_{t-1}$ | −0.192 | −0.133 | −0.142 | −0.228 | −0.265 | −0.179 | −0.210 | −0.149 | −0.188 |
| | (0.690) | (0.691) | (0.691) | (0.687) | (0.688) | (0.688) | (0.689) | (0.688) | (0.689) |
| Slack resources$_{t-1}$ | −0.390 | −0.314 | −0.455 | −0.395 | −0.400 | −0.331 | −0.459 | −0.318 | −0.468 |
| | (0.488) | (0.487) | (0.486) | (0.489) | (0.489) | (0.488) | (0.486) | (0.488) | (0.486) |
| R and D investment$_{t-1}$ | −6.223 | −8.040 | −6.255 | −6.758 | −6.675 | −8.279 | −6.809 | −7.757 | −5.949 |
| | (9.017) | (8.970) | (9.027) | (9.047) | (9.034) | (9.001) | (9.050) | (8.982) | (9.065) |
| Inverse mills ratio$_{t-1}$ | −1.385 ** | −1.404 ** | −1.172 * | −1.307 ** | −1.359 ** | −1.325 ** | −1.133 * | −1.331 ** | −1.122 * |
| | (0.458) | (0.463) | (0.457) | (0.456) | (0.459) | (0.460) | (0.458) | (0.458) | (0.456) |
| $I_1(P_{t-1} - A_{t-1}) < 0$ | | −20.89 *** | −51.20 * | | | −17.48 ** | −60.19 ** | −20.28 *** | −57.64 ** |
| | | (6.149) | (20.11) | | | (6.125) | (20.14) | (6.154) | (20.17) |
| $I_2(EP_{t+1} - T_t) < 0$ | | | | 185.6 *** | 111.2 *** | 180.4 *** | 122.9 *** | 179.2 *** | 152.7 *** |
| | | | | (26.27) | (21.24) | (26.43) | (21.19) | (26.51) | (24.12) |
| $I_1(P_{t-1} - A_{t-1}) I_2(EP_{t+1} - T_t)$ | | | | | | | | 2439.3 *** | 1280.6 ** |
| | | | | | | | | (694.3) | (488.0) |
| $(1 - I_1)(P_{t-1} - A_{t-1}) \geq 0$ | | −6.850 | −15.97 | | | −5.552 | −21.98 | −6.374 | −23.45 |
| | | (6.925) | (21.26) | | | (6.858) | (21.14) | (6.822) | (21.11) |
| $(1 - I_2)(EP_{t+1} - T_t) \geq 0$ | | | | 0.949 | 2.874 | 0.677 | 0.415 | 0.636 | 0.288 |
| | | | | (3.440) | (3.466) | (3.442) | (3.420) | (3.425) | (3.403) |
| Year effect | Y | Y | Y | Y | Y | Y | Y | Y | Y |
| Constants | 30.00 *** | 30.20 *** | 30.46 *** | 29.84 *** | 30.09 *** | 30.01 *** | 30.67 *** | 30.01 *** | 30.55 *** |
| | (1.170) | (1.171) | (1.187) | (1.157) | (1.163) | (1.159) | (1.179) | (1.157) | (1.182) |
| N | 10,280 | 10,279 | 10,280 | 10,280 | 10,280 | 10,279 | 10,280 | 10,279 | 10,280 |
| $R^2$ | 0.147 | 0.148 | 0.149 | 0.154 | 0.150 | 0.155 | 0.152 | 0.156 | 0.153 |
| F | 40.28 *** | 36.56 *** | 36.90 *** | 38.03 *** | 37.41 *** | 34.80 *** | 34.71 *** | 33.66 *** | 33.33 *** |

Notes: † $p < 0.1$, * $p < 0.05$, ** $p < 0.01$, *** $p < 0.001$. Robust standard errors in parentheses.

**Table 4.** Results of Moderating Effects.

| | Model 5 (Backward-Looking) | | Model 6 (Forward-Looking) | | Model 7 (Joint Model) | |
|---|---|---|---|---|---|---|
| | Firm | Industry | Firm | Industry | Firm | Industry |
| $CSR_{t-1}$ | 0.0740 *** | 0.0730 *** | 0.0710 *** | 0.0743 *** | 0.0698 *** | 0.0723 *** |
| | (0.0161) | (0.0162) | (0.0160) | (0.0160) | (0.0161) | (0.0162) |
| Median of Industry $CSR_t$ | 0.468 *** | 0.455 *** | 0.446 *** | 0.477 *** | 0.436 *** | 0.447 *** |
| | (0.0701) | (0.0698) | (0.0702) | (0.0707) | (0.0695) | (0.0695) |
| Boardroom gender diversity$_{t-1}$ | −4.760 † | −4.514 | −4.348 | −4.740 † | −4.490 | −4.630 |
| | (2.843) | (2.838) | (2.834) | (2.839) | (2.835) | (2.834) |
| Subsidy$_{t-1}$ | 0.0548 | 0.0568 | 0.0518 | 0.0626 | 0.0533 | 0.0614 |
| | (0.133) | (0.134) | (0.132) | (0.134) | (0.132) | (0.134) |
| Ownership type$_{t-1}$ | −2.036 † | −1.929 | −2.223 † | −2.030 † | −2.346 † | −2.062† |
| | (1.231) | (1.221) | (1.227) | (1.231) | (1.227) | (1.221) |
| Visibility$_{t-1}$ | 14.20 * | 12.00 † | 18.50 ** | 17.71 ** | 15.11 * | 12.06† |
| | (6.781) | (6.724) | (6.740) | (6.801) | (6.762) | (6.731) |
| $ROA_{t-1}$ | 42.43 *** | 59.39 ** | 18.62 *** | 18.19 *** | 36.20 *** | 63.18 ** |
| | (7.431) | (20.47) | (4.806) | (4.810) | (7.398) | (20.46) |
| Firm age$_{t-1}$ | −12.08 | −12.74 | −14.82 | −15.29 | −11.09 | −11.29 |
| | (13.29) | (13.26) | (13.06) | (13.13) | (13.17) | (13.18) |
| Firm size$_{t-1}$ | 175.1 *** | 184.6 *** | 183.5 *** | 184.4 *** | 177.9 *** | 187.0 *** |
| | (29.69) | (29.70) | (28.99) | (29.32) | (29.35) | (29.66) |
| Political ties$_{t-1}$ | −0.158 | −0.167 | −0.238 | −0.248 | −0.206 | −0.214 |
| | (0.689) | (0.690) | (0.686) | (0.688) | (0.687) | (0.689) |
| Slack resources$_{t-1}$ | −0.425 | −0.544 | −0.387 | −0.420 | −0.415 | −0.534 |
| | (0.487) | (0.486) | (0.488) | (0.488) | (0.487) | (0.485) |
| R and D investment$_{t-1}$ | −7.717 | −6.051 | −6.068 | −6.165 | −7.475 | −6.262 |
| | (8.981) | (8.974) | (9.043) | (9.049) | (9.013) | (9.009) |
| Inverse mills ratio$_{t-1}$ | −1.245 ** | −0.914 * | −1.293 ** | −1.326 ** | −1.185 * | −0.924 * |
| | (0.468) | (0.450) | (0.458) | (0.456) | (0.466) | (0.451) |
| $I_1(P_{t-1} - A_{t-1}) < 0$ | −9.771 | 3.650 | | | −6.590 | −12.46 |
| | (6.376) | (22.14) | | | (6.275) | (22.25) |
| $ROA_{t-1}I_1(P_{t-1} - A_{t-1})$ | 200.7 *** | 400.9 *** | | | 172.9 ** | 331.7 *** |
| | (56.89) | (84.59) | | | (56.42) | (84.77) |
| $I_2(EP_{t+1} - T_t) < 0$ | | | 243.8 *** | 148.3 *** | 228.6 *** | 128.0 *** |
| | | | (32.09) | (28.55) | (31.97) | (27.85) |
| $ROA_{t-1}I_2(EP_{t+1} - T_t)$ | | | 1573.9 ** | 958.3 * | 1367.6 ** | 544.0 |
| | | | (527.2) | (482.9) | (517.5) | (474.6) |
| $(1 - I_1)(P_{t-1} - A_{t-1}) \geq 0$ | −12.52 † | −33.53 | | | −10.28 | −37.21† |
| | (6.997) | (21.78) | | | (6.923) | (21.55) |
| $(1 - I_2)(EP_{t+1} - T_t) \geq 0$ | | | −0.372 | 2.201 | −2.659 | −0.840 |
| | | | (3.475) | (3.462) | (3.431) | (3.428) |
| Year effect | Y | Y | Y | Y | Y | Y |
| Constants | 30.12 *** | 29.82 *** | 29.55 *** | 29.96 *** | 29.70 *** | 30.05 *** |
| | (1.171) | (1.194) | (1.162) | (1.167) | (1.164) | (1.193) |
| N | 10,279 | 10,280 | 10,280 | 10,280 | 10,279 | 10,280 |
| $R^2$ | 0.150 | 0.152 | 0.155 | 0.151 | 0.157 | 0.155 |
| F | 35.43 | 35.85 | 36.30 | 35.66 | 32.39 | 32.20 |

Notes: † $p < 0.1$, * $p < 0.05$, ** $p < 0.01$, *** $p < 0.001$. Robust standard errors in parentheses.

Model 1 of Table 3 shows the impacts on CSR in the backward-looking pattern. The negative and significant coefficients of $(I_1(P_{t-1} - A_{t-1}) < 0)$ indicate that CSR increases with the distance by which financial performance in the last year falls below performance goals (firm- or industry-specific). It is consistent with arguments that underperformance will motivate organizations to search for alternative plans [37,56] and losses in the past will make firms more risk-averse [22]. Thus hypothesis 1 is supported. Model 2 of Table 3 reports the impacts on CSR in the forward-looking pattern. The positive and significant coefficients of $(I_2(EP_{t+1} - T_t) < 0)$ indicate that CSR decreases with the distance by which expected financial performance in the next year will fall below performance targets (firm- or industry-specific). Therefore, hypothesis 2 is supported. It echoes the ideas from the behavioral theory of firm that underperformance will motivate organizations to search for alternative plans [6,37] and ideas from prospect theory that likely losses in future will make organizations more risk-seeking [22].

### 4.3. Interaction Between Backward and Forward Search

Model 3 presents the joint impact of both backward and forward-looking performance gaps on CSR. We found that the sign and significance of both $(I_1(P_{t-1} - A_{t-1}) < 0)$ and $(I_2(EP_{t+1} - T_t) < 0)$ remains unchanged. Model 4 from Table 3 reports the results of the moderating effect of the expected performance gap in the next year on the relationship between the performance gap in the last year and CSR in this year. The coefficients of moderation term are positive and significant which indicates that expected loss in future will increase organizational tolerance to risk and thus seek for more risk-taking alternative plans, such as R and D actions. Then hypothesis 3 is supported.

### 4.4. Moderating Effect of Value of Financial Performance

Table 4 mainly presents the moderating effect of the value of financial performance in the last year on the impacts of different determinants on CSR. Model 5 reports the moderating effect of the value of financial performance on the relationship between the performance gap in the last year and CSR in this year. The results show that the coefficients of the performance gap in the last year become insignificant after including an interaction term, which indicates that the interaction term should have a more powerful impact on CSR. This is an interesting finding since prior studies mostly focused on the value of financial performance [63] or performance gap [6] respectively, and few of them have tried to investigate their interactive effect. The positive and significant coefficients of the interaction term indicate that firms will have less motivation to conduct CSR practices if they think their financial performance in last year alone is good enough. Rationally speaking, performance gap is a better reference to make decisions [64]. But we have to admit that the value of financial performance also influences organizational decisions and behaviors. Thus hypothesis 4 is supported.

Model 6 from Table 4 presents the results of the moderating effect of the value of financial performance in the last year on the effect of expected performance gap against targets. The positive and significant coefficients of interaction terms show that firms will have less motivation to invest in CSR practices if the value of financial performance in the last year is good enough and they are going to lose in the next year. Again, this result is supportive of our arguments that the mixed findings of the value of financial performance on organizational decisions and behaviors [63,65] does not mean that it does not make sense, but the way it makes sense to firms should be as an effective boundary. Thus hypothesis 5 is supported.

Model 7 included both of these two moderating effects and got similar results. These two moderating roles suggest that the impacts of performance gap and the value of financial performance could coexist to influence organizational decisions. It is better to take both of them into consideration to have a better understanding of corporate decisions and behaviors. It will make results biased and misconducted if we only focus on one of these two perspectives.

*4.5. Figures*

We used Figures 1–3 to better understand the main effect and these moderating effects based on estimated coefficients in firm aspiration/target regression. Figure 1 presents the main effect of the backward-looking performance gap on CSR as well as the moderating effect of the forward-looking performance gap. The bold line displays the main effect of the backward-looking performance gap which has been hypothesized in hypothesis 1, and the slope shows that CSR increases with the distance by which financial performance in the last year falls below the goals. The two dotted lines are plotted assuming the value of the forward-looking performance gap is the mean plus or minus one standard deviation. These three lines have some conjunctions, but we could see that the slope of the line with a high forward-looking performance gap is flatter than lines with a neutral or low forward-looking performance gap. That is to say, the expected performance gap in the next year will weaken the effect of the performance gap in the last year on CSR.

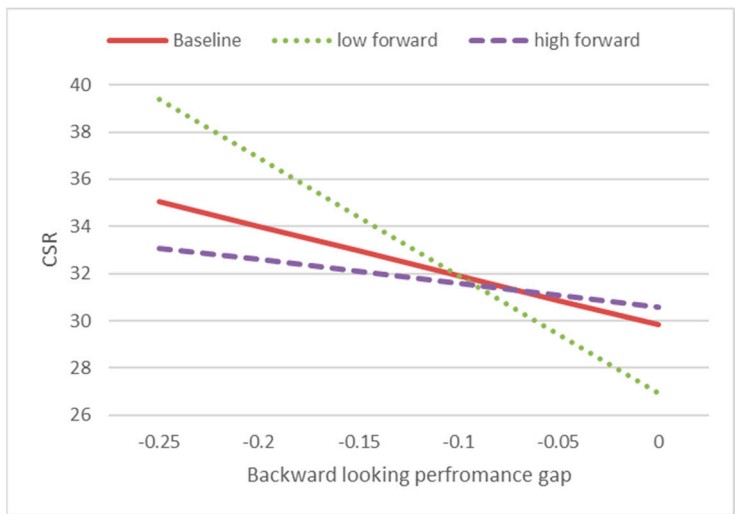

**Figure 1.** Moderating effect of forward performance gap on relationship between backward performance gap and corporate social responsibility (CSR).

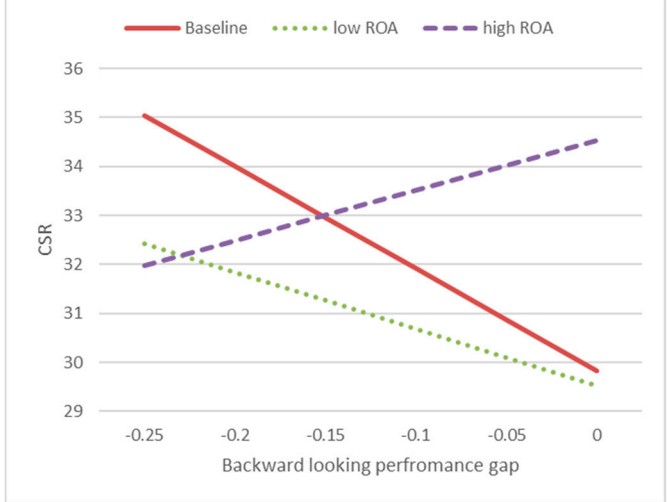

**Figure 2.** Moderating effect of the value of financial performance on the relationship between the backward performance gap and CSR.

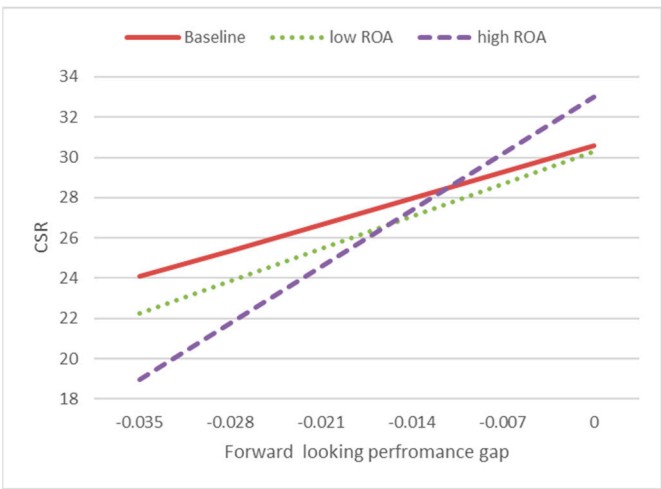

**Figure 3.** Moderating effect of the value of financial performance on the relationship between the forward performance gap and CSR.

Figure 2 presents the main effect of the backward-looking performance gap on CSR as well as the moderating effect of the value of financial performance in the last year. The two dotted lines are plotted assuming the value of the moderator is the mean plus or minus one standard deviation. We can see that the line with low ROA in the last year has a flatter slope than the line with neutral ROA in the last year. It is interesting to find that CSR would decrease with the distance by which financial performance falls below goals in the last year if the value of financial performance is mean plus one standard deviation. It is a powerful support to our argument that the value of financial performance has a moderating effect on the effect of the relative value of financial performance against goals on CSR.

Figure 3 depicts the main effect of the forward-looking performance gap on CSR as well as the moderating effect of the value of financial performance in the last year. The bold line displays the main effect and we can see that CSR decreases with the distance by which expected performance will fall below performance targets as we have hypothesized in hypothesis 2. The two dotted lines are plotted assuming the value of the moderator is the mean plus or minus one standard deviation. CSR would decrease at a higher rate with the distance by which expected financial performance will fall below performance targets if the value of financial performance in the last year is good enough.

*4.6. Robustness Examination*

We first used the Generalized Least Squares (GLS) model to retest our hypotheses to avoid bias from the regression method. Estimation from the GLS model is unbiased from the existence of AR (1) autocorrelation, cross-sectional correlation, and heteroskedasticity across panels. The results of the main effects with the GLS method are reported in Table 5. We can see that our hypotheses are still supported by the data.

Squared terms of independents were added into regressions to see if the curvilinear relationship can better describe the backward- and forward-looking CSR activities. Low and moderate distance below performance targets or goals may provide motivation to search for alternative plans, but high distance below performance targets or goals may limit corporate ability to do so [66]. The results are reported in Table 6. We can see that most squared items are not significant and the $R^2$ values are not larger than linear models, which indicate that the linear relationship is a better description.

**Table 5.** Results of Main Effect with the Generalized Least Squares (GLS) Model.

| | Model 8 (Backward-Looking) | | Model 9 (Forward-Looking) | | Model 10 (Joint Model) | |
|---|---|---|---|---|---|---|
| | Firm | Industry | Firm | Industry | Firm | Industry |
| $CSR_{t-1}$ | 0.677 *** | 0.695 *** | 0.679 *** | 0.690 *** | 0.677 *** | 0.684 *** |
| | (0.00710) | (0.00668) | (0.00752) | (0.00731) | (0.00749) | (0.00628) |
| Median of Industry $CSR_t$ | 0.215 *** | 0.243 *** | 0.141 *** | 0.183 *** | 0.209 *** | 0.223 *** |
| | (0.0208) | (0.0201) | (0.0196) | (0.0200) | (0.0214) | (0.0206) |
| Boardroom gender diversity$_{t-1}$ | 1.349 ** | 0.308 | 1.973 *** | 2.041 *** | 0.842 | 0.0454 |
| | (0.507) | (0.505) | (0.478) | (0.498) | (0.513) | (0.483) |
| Subsidy$_{t-1}$ | −0.0612 | −0.0559 | 0.0107 | 0.00562 | −0.0116 | 0.0490 |
| | (0.0411) | (0.0376) | (0.0439) | (0.0444) | (0.0422) | (0.0391) |
| Ownership type$_{t-1}$ | −0.0607 | 0.0186 | −0.331 * | −0.172 | −0.238 | −0.214 |
| | (0.141) | (0.139) | (0.143) | (0.139) | (0.145) | (0.130) |
| Visibility$_{t-1}$ | 4.339 *** | 4.462 *** | 5.013 *** | 6.314 *** | 3.630 *** | 3.680 *** |
| | (0.966) | (0.906) | (0.936) | (0.956) | (0.967) | (0.940) |
| $ROA_{t-1}$ | 20.38 *** | 21.02 *** | 0.0643 | 3.502 * | 8.924 *** | 13.05 *** |
| | (1.525) | (3.893) | (1.587) | (1.590) | (1.800) | (3.605) |
| Firm age$_{t-1}$ | 2.082 *** | 1.710 *** | 2.806 *** | 2.713 *** | 2.258 *** | 1.416 *** |
| | (0.418) | (0.444) | (0.453) | (0.457) | (0.480) | (0.380) |
| Firm size$_{t-1}$ | 45.10 *** | 49.69 *** | 69.38 *** | 53.55 *** | 61.27 *** | 57.64 *** |
| | (4.278) | (4.164) | (4.345) | (4.255) | (4.523) | (4.079) |
| Political ties$_{t-1}$ | −0.155 | 0.315 * | 0.0643 | 0.0229 | −0.0981 | 0.0266 |
| | (0.140) | (0.129) | (0.124) | (0.139) | (0.144) | (0.127) |
| Slack resources$_{t-1}$ | 0.158 | −0.404 * | −0.328 | −0.404 | 0.210 | 0.0179 |
| | (0.191) | (0.198) | (0.205) | (0.213) | (0.189) | (0.178) |
| R and D investment$_{t-1}$ | 0.819 | 1.590 | 6.074 *** | 4.926 ** | 0.538 | −1.594 |
| | (1.585) | (1.702) | (1.714) | (1.698) | (1.699) | (1.740) |
| Inverse mills ratio$_{t-1}$ | −1.422 *** | −1.394 *** | −1.078 *** | −1.506 *** | −1.265 *** | −1.133 *** |
| | (0.251) | (0.157) | (0.224) | (0.184) | (0.230) | (0.198) |
| $I_1(P_{t-1} - A_{t-1}) < 0$ | −20.90 *** | −29.71 *** | | | −17.71 *** | −27.20 *** |
| | (2.315) | (4.220) | | | (2.253) | (4.019) |
| $I_2(EP_{t+1} - T_t) < 0$ | | | 75.93 *** | 75.52 *** | 111.5 *** | 118.9 *** |
| | | | (8.611) | (7.826) | (9.276) | (7.988) |
| $(1 - I_1)(P_{t-1} - A_{t-1}) \geq 0$ | −7.218 ** | 4.842 | | | −4.012 | 7.669 |
| | (2.635) | (4.308) | | | (2.561) | (4.113) |
| $(1 - I_2)(EP_{t+1} - T_t) \geq 0$ | | | 11.29 *** | 6.256 *** | 5.125 *** | −0.317 |
| | | | (0.741) | (0.773) | (1.014) | (0.994) |
| Year effect | Y | Y | Y | Y | Y | Y |
| Constants | 27.77 *** | 28.12 *** | 29.33 *** | 27.97 *** | 28.01 *** | 29.37 *** |
| | (0.183) | (0.175) | (0.134) | (0.196) | (0.189) | (0.137) |
| N | 10,279 | 10,280 | 10,280 | 10,280 | 10,279 | 10,280 |
| Wald Chi2 | 21,647.39 | 21,329.57 | 47,472.18 | 20,195.46 | 20,413.93 | 31,928.89 |

Notes: * $p < 0.05$, ** $p < 0.01$, *** $p < 0.001$. Standard errors in parentheses.

**Table 6.** Results of Curvilinear Relationship between Independents and Dependent.

| | Model11 (Backward-looking) | | Model12 (Forward-looking) | | Model13 (Joint Model) | |
|---|---|---|---|---|---|---|
| | Firm | Industry | Firm | Firm | Industry | Firm |
| $CSR_{t-1}$ | 0.0730 *** | 0.0745 *** | 0.0692 *** | 0.0728 *** | 0.0673 *** | 0.0732 *** |
| | (0.0162) | (0.0162) | (0.0159) | (0.0160) | (0.0161) | (0.0162) |
| Median of Industry $CSR_t$ | 0.478 *** | 0.459 *** | 0.450 *** | 0.473 *** | 0.446 *** | 0.447 *** |
| | (0.0707) | (0.0700) | (0.0703) | (0.0709) | (0.0700) | (0.0697) |
| Boardroom gender diversity$_{t-1}$ | −4.791 | −4.507 | −4.406 | −4.757 | −4.615 | −4.671 |
| | (2.841) | (2.843) | (2.841) | (2.845) | (2.839) | (2.842) |
| Subsidy$_{t-1}$ | 0.0418 | 0.0601 | 0.0593 | 0.0607 | 0.0502 | 0.0660 |
| | (0.133) | (0.134) | (0.133) | (0.134) | (0.132) | (0.134) |
| Ownership type$_{t-1}$ | −2.023 | −1.930 | −2.076 | −1.968 | −2.239 | −2.051 |
| | (1.225) | (1.231) | (1.233) | (1.233) | (1.227) | (1.231) |
| Visibility$_{t-1}$ | 17.20 * | 12.59 | 17.91 ** | 18.12 ** | 17.06 * | 12.52 |
| | (6.805) | (6.754) | (6.762) | (6.801) | (6.752) | (6.749) |
| $ROA_{t-1}$ | 34.73 *** | 57.46 ** | 20.43 *** | 18.89 *** | 31.66 *** | 61.54 ** |
| | (7.251) | (20.42) | (4.835) | (4.843) | (7.233) | (20.38) |
| Firm age$_{t-1}$ | −13.93 | −13.42 | −14.06 | −15.46 | −11.80 | −12.04 |
| | (13.31) | (13.26) | (13.05) | (13.12) | (13.17) | (13.17) |
| Firm size$_{t-1}$ | 176.7 *** | 184.1 *** | 184.3 *** | 186.1 *** | 180.4 *** | 188.5 *** |
| | (29.61) | (29.70) | (29.10) | (29.33) | (29.37) | (29.64) |
| Political ties$_{t-1}$ | −0.127 | −0.162 | −0.238 | −0.249 | −0.182 | −0.206 |
| | (0.691) | (0.690) | (0.687) | (0.687) | (0.689) | (0.689) |
| Slack resources$_{t-1}$ | −0.338 | −0.520 | −0.402 | −0.390 | −0.356 | −0.499 |
| | (0.488) | (0.486) | (0.490) | (0.489) | (0.489) | (0.486) |
| R and D investment$_{t-1}$ | −7.580 | −6.281 | −6.744 | −6.588 | −7.934 | −6.763 |
| | (9.004) | (9.002) | (9.059) | (9.013) | (9.042) | (9.010) |
| Inverse mills ratio$_{t-1}$ | −1.382 ** | −1.002 * | −1.316 ** | −1.343 ** | −1.319 ** | −0.989 * |
| | (0.466) | (0.454) | (0.456) | (0.457) | (0.463) | (0.453) |
| Squared $I_1(P_{t-1} - A_{t-1}) < 0$ | 86.81 | 281.9 *** | | | 69.25 | 219.5 ** |
| | (57.40) | (82.52) | | | (57.06) | (83.00) |
| $I_1(P_{t-1} - A_{t-1}) < 0$ | −10.00 | −17.52 | | | −8.855 | −33.05 |
| | (9.486) | (21.67) | | | (9.423) | (21.66) |
| Squared $I_2(EP_{t+1} - T_t) < 0$ | | | −2386.8 | 3065.2 | −2511.5 | 2775.7 |
| | | | (2660.4) | (1623.8) | (2659.4) | (1620.0) |
| $I_2(EP_{t+1} - T_t) < 0$ | | | 141.8 ** | 185.2 *** | 133.5 * | 181.2 *** |
| | | | (55.00) | (46.38) | (55.02) | (46.21) |
| $(1 - I_1)(P_{t-1} - A_{t-1}) \geq 0$ | −7.670 | −21.98 | | | −6.299 | −25.66 |
| | (6.888) | (21.41) | | | (6.825) | (21.27) |
| $(1 - I_2)(EP_{t+1} - T_t) \geq 0$ | | | 1.318 | 2.158 | 0.696 | −0.859 |
| | | | (3.468) | (3.481) | (3.474) | (3.435) |
| Year effect | Y | Y | Y | Y | Y | Y |
| Constants | 30.03 *** | 30.05 *** | 30.01 *** | 29.84 *** | 30.07 *** | 30.11 *** |
| | (1.183) | (1.195) | (1.167) | (1.171) | (1.181) | (1.196) |
| N | 10,279 | 10,280 | 10,280 | 10,280 | 10,279 | 10,280 |
| $R^2$ | 0.148 | 0.150 | 0.154 | 0.150 | 0.155 | 0.154 |
| F | 34.88 | 35.58 | 36.31 | 35.73 | 32.04 | 32.11 |

Notes: * $p < 0.05$, ** $p < 0.01$, *** $p < 0.001$. Robust standard errors in parentheses.

## 5. Discussion and Conclusion

### 5.1. Conclusions

Based on the behavioral theory of firm and prospect theory, we tried to investigate how CSR activities are influenced by financial performance. Prior studies on this relationship have led to mixed findings since they assumed that it is the value of financial performance that determines CSR activities. To address such mixed findings, this study assumes that it is the performance gaps against goals or targets that determines CSR activities due to the existence of bounded rationality. Besides, we found that underperformance in the past is more likely to encourage firms to engage in less risk-taking behaviors such as CSR activities, and likely underperformance in the future would encourage firms to engage in less CSR practices. Finally, this paper found that the value of financial performance is such an important boundary on the relationships in question that the impact of past underperformance would even be positive under the high value of the financial performance situation.

### 5.2. Theory Contributions

Our findings extend the literature on organization research in three ways. First, we contributed to CSR literature by finding its determinant as underperformance gaps. Even though there are dozens of studies on determinants of CSR activities, such as corporate governance [67], firm size, and competition intensity [4], little is known about the relationship between financial performance gap and CSR decisions. The inconsistent prior findings on the relationship between financial performance and CSR activities may result from their negligence of organizational bounded rationality. Specifically, we argued that organizations would set goals or targets as a reference point to evaluate their financial performance [6] and it is the performance gaps between real financial performance and these reference points that determines CSR activities. It is a new perspective and can help gain better understanding on the determinants of CSR activities.

Second, we extend literature on the behavioral theory of firm and organizational risk-taking by providing CSR as a less risk-taking choice. Prior studies about performance gap and organizational behaviors mostly focused on high risk-seeking actions, such as R and D investment [6]. We provide CSR as an alternative choice which is less risk-taking and thus, present a more completed picture. Though it has been predicted by prospect theory that losses in the past will make organizations more risk-averse and losses in the future will make organizations more risk-seeking [22], most studies have tested this logic only by high risk-taking activities such as R and D actions. Prior findings that firms would increase R and D investments when facing underperformance in the last year [6] actually contradicts the predictions of prospect theory. One possible explanation may be that these studies did not take the trade-off between CSR and R and D activities into consideration. This paper did control the R and D investment when exploring CSR activities and got results consistent with prospect theory. Therefore, future studies on CSR or R and D should take organizational trade-off between them into consideration.

Last but not least, we shed new light on the behavioral theory of firm by incorporating the value of financial performance as an important boundary. Prior studies have investigated the effects of performance gaps and the value of financial performance, respectively [6,65]. Mostly, they thought that performance gap was a better determinant of organizational behaviors than the value of financial performance alone [41,56] due to the existence of bounded rationality, implicitly or explicitly. We argued that the value of financial performance will influence the impacts of performance gaps on corporate decisions, which may be another form of bounded rationality. This study found that the value of financial performance will change the effect of underperformance gaps and increase organizational tolerance to risk. In this way, incorporating these two perspectives into a comprehensive framework and investigating their interactive effects is helpful to gain better understandings.

### 5.3. Implications for Practitioners

The results obtained in this paper have some implications for practitioners. First, managers could choose CSR as alternative plan when facing past failures. Great pressures are faced by managers when firms fail to attain performance goals, and managers need some effective alternatives to respond to these pressures [28]. Prior studies may encourage firms to promote investment into R and D activities, for such behaviors can build competition advantage [6]. However, managers may be less likely to take such suggestions since failure in the past makes them more risk-averse [22]. We suggest managers in such a situation promote investment into CSR activities, for such behaviors can build competition advantage [36] and satisfy managers' needs of low risk-taking at the same time.

Second, managers could choose high risk-taking alternative plans when there is a possibility to fail expectations in the future. In line with prior studies, results from this paper suggest managers increase investment into R and D activities if firms are unable to reach performance targets in the future [6]. In such a situation, managers should invest more in high risk-taking behaviors, such as R and D, to avoid possible failure in the future.

Third, managers should take both performance gaps in past and in future into consideration when making decisions. Both history experience and future expectations are useful and important reference points in the decision-making process. Thus, managers should take both backward- and forward-looking performance gaps into consideration when looking for effective alternative plans. Focusing on only one of these two perspectives may lead to unsatisfactory consideration of the need from another perspective.

Finally, managers should avoid the illusion of the value of financial performance. Judgments of business operation may be influenced by the value of financial performance even though firms failed, or are going to fail, reference points. However, reference points are set to cope with bounded rationality and the impact of the value of financial performance may distract firms far away from rationality. In this way, managers should make judgements and decisions based on performance gaps and avoid the illusion of the value of financial performance.

### 5.4. Limitations and Future Study Considerations

However, there are some limitations in our studies. First, we did not incorporate firms' choice among high risk-taking behaviors such as R and D actions and less risk-taking behaviors such as CSR practices into a framework. The effect of performance gap on R and D actions has been investigated in a US setting [1] and we assume that there will be consistent findings in Chinese settings during our hypotheses. This assumption needs further investigations and confirmations. Second, CSR activities can be conducted in different domains [68] and different combinations of these domains may have different meanings to firms [69,70]. We only discussed the impacts of performance gaps on overall CSR activities in this paper. Future studies could try to investigate how firms will behave in these domains under different performance gaps situations. Third, we did not investigate the effect of outperformance on CSR. We argued that CSR could be used to solve problems since underperformance firms will have motivations to search for alternative plans. But it is likely that outperformance firms also have motivations to promote CSR to build competitive advantage. It is another story and future studies hold promise to have promising findings.

**Author Contributions:** Conceptualization, X.D.; methodology, X.D. and X.L.; formal analysis, X.L.; investigation, X.L.; resources, X.D.; data curation, X.L.; writing—original draft preparation, X.D. and X.L.; writing—review and editing, X.D. and X.L.; supervision, X.D.; project administration, X.D.; funding acquisition, X.D.

**Funding:** This research was funded by National Natural Science Foundation of China (71872132, 71572132); The Young Scholars' Academic Development Program of Wuhan University's Humanities and Social Sciences (Whu 2016011).

**Conflicts of Interest:** The authors declare no conflict of interest.

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
