# Peer review of "Financial Performance Gaps and Corporate Social Responsibility"

_sustainability, doi:10.3390/su11123438_

Round 1

Reviewer 1 Report

This paper investigates how CSR responds to different performance gaps, for a panel dataset of Chinese listed firms from 2011 to 2016,

I have enjoyed reading the paper, and indeed there are no major changes I would suggest. Only a few minor issues:

Equation (1) is apparently not numbered.

Congratulations for your effort

Reviewer 2 Report

The paper is well structured even if it could be published also in a standard 10 pages version with some cuts that won't change its value and this is a good point in general.

I think it is good to be published in this form.

Reviewer 3 Report

Many thanks for the opportunity to read and review the paper "Financial performance gaps and corporate social responsibility"

I would suggest the authors to change the structure of the introduction section and structure it into four paragraphs focusing on: what we know, what we do not know, how we get to know (theory and empirics) and why should we care (contributions). This would make easier for the reader to understand the actual hook of the paper as well as contextualize better the purpose of the study.

suggest incorporating these papers into your theoretical background, which may also provide with clarity for developing your theoretical implications:

You may want to give a look:

Cavaco, S., & Crifo, P. (2014). CSR and financial performance: complementarity between environmental, social and business behaviours. Applied Economics46(27), 3323-3338.

Crifo, P., Diaye, M. A., & Pekovic, S. (2016). CSR related management practices and firm performance: An empirical analysis of the quantity–quality trade-off on French data. International Journal of Production Economics171, 405-416.

Lagasio, V., & Cucari, N. (2018). Corporate governance and environmental social governance disclosure: A meta‐analytical review. Corporate Social Responsibility and Environmental Management.

Manrique, S., & Martí-Ballester, C. P. (2017). Analyzing the effect of corporate environmental performance on corporate financial performance in developed and developing countries. Sustainability9(11), 1957.

Crifo, P., Escrig-Olmedo, E., & Mottis, N. (2018). Corporate governance as a key driver of corporate sustainability in France: The role of board members and investor relations. Journal of Business Ethics, 1-20.

In the Conclusion section, please avoid repetitions here and only highlight the importance of the topic and the main implications of your results. In other words, Results do not have to be summarized again, only (the very interesting!) contributions should remain.

Minor: 

I suggest doing the abstract more attractive

I hope that these recommendations are helpful to you. Good luck!

Author Response

please see the attached file as follows
